# The Roles of MicroRNA in Lung Cancer

**DOI:** 10.3390/ijms20071611

**Published:** 2019-03-31

**Authors:** Kuan-Li Wu, Ying-Ming Tsai, Chi-Tun Lien, Po-Lin Kuo, Jen-Yu Hung

**Affiliations:** 1Division of Pulmonary and Critical Care Medicine, Department of Internal Medicine, Kaohsiung Medical University Hospital, Kaohsiung Medical University, Kaohsiung 807, Taiwan, 980448kmuh@gmail.com (K.-L.W.); tsaiyingming@gmail.com (Y.-M.T.); 2Department of Internal Medicine, Kaohsiung Municipal Ta-Tung Hospital, Kaohsiung Medical University, Kaohsiung 807, Taiwan; 3Graduate Institute of Medicine, College of Medicine, Kaohsiung Medical University, Kaohsiung 807, Taiwan; 4Department of Internal Medicine, School of Medicine, College of Medicine, Kaohsiung Medical University, Kaohsiung 807, Taiwan; 5Department of Respiratory Care, College of Medicine, Kaohsiung Medical University, Kaohsiung 807, Taiwan; 6Department of Internal Medicine, Kaohsiung Municipal United Hospital, Kaohsiung 804, Taiwan; kmuha10537@gmail.com; 7Graduate Institute of Clinical Medicine, College of Medicine, Kaohsiung Medical University, Kaohsiung 807, Taiwan; kuopolin@seed.net.tw

**Keywords:** microRNA, lung cancer

## Abstract

Lung cancer is the most devastating malignancy in the world. Beyond genetic research, epigenomic studies—especially investigations of microRNAs—have grown rapidly in quantity and quality in the past decade. This has enriched our understanding about basic cancer biology and lit up the opportunities for potential therapeutic development. In this review, we summarize the involvement of microRNAs in lung cancer carcinogenesis and behavior, by illustrating the relationship to each cancer hallmark capability, and in addition, we briefly describe the clinical applications of microRNAs in lung cancer diagnosis and prognosis. Finally, we discuss the potential therapeutic use of microRNAs in lung cancer.

## 1. Introduction

Lung cancer is the leading cause of cancer deaths worldwide. In the GLOBOCAN 2018 database, 2.09 million new cases and 1.76 million deaths from lung cancers are estimated [1]. Histologically, lung cancer can be divided into small-cell lung cancer (SCLC) and non-small-cell lung cancer (NSCLC). The latter comprises more than 80% of lung cancer cases and is subdivided into adenocarcinoma, squamous cell carcinoma, and large-cell carcinoma [2]. The development of lung cancer is complex, being a result of genetic and environmental interactions. Risk factors include tobacco consumption mainly in the form of cigarette smoking, radiation exposure, and environmental toxins, e.g., radon, asbestos, arsenic, chromium, and nickel, amongst others. In 2010, the National Lung Screen Trial (NLST) reported a 20% reduction of lung cancer mortality in the high-risk population by using low-dose computed tomography (LDCT) for screening purposes. A major concern with the LDCT screening method is its high false-positive rate, which leads to overdiagnosis, unnecessary radiation exposure, economic burden, and patient anxiety [3]. Thus, the development of more accurate methods for the diagnosis of lung cancer is urgently needed. Despite emerging advances in early diagnosis and novel targeted agents, the prognosis of lung cancer remains poor. Patients with lung cancer are often diagnosed at advanced stages, and usually more than 60% of patients are at stages III or IV before treatment [4]. Therefore, the overall 5-year survival rate of lung cancer remains low at 16.8%, and is less than 5% among those with metastatic disease [5]. To improve diagnosis of lung cancer, we still need to develop a thorough understanding of its tumorigenesis, not only from the aspect of genetics, but also epigenetics.

MicroRNAs (miRNA) are a family of small noncoding RNAs (21–25 nucleotides) that can inhibit messenger RNA (mRNA) translation and promote mRNA degradation by base pairing to complementary sites of target mRNAs. Through this mechanism, miRNAs alter gene expression post-transcriptionally. The first noncoding RNA, *lin-4*, was identified as miRNA in *Caenorhabditis elegans* in 1993 [6]; however, miRNAs were not defined as a distinct group of regulatory molecules until the early 2000s, and thereafter, extensive studies about miRNAs grew rapidly in number, while now, over 2500 mature miRNAs (from 1188 miRNA precursors) have been identified in the miRBase database [7]. The biogenesis of miRNAs from miRNA genomic loci is a multistep process. The miRNA precursors, pri-miRNAs, are large miRNAs (>100 nucleotides in length) transcribed by RNA polymerase II and subsequently processed, intranuclearly, by the RNase III enzyme, Drosha, and the double-stranded RNA (dsRNA)-binding protein, Pasha (also known as DiGeorge Syndrome critical region gene 8, DGCR8) [8]. The product of this process is called pre-miRNA, with a length of ~70 nucleotides. The pre-miRNAs are then transported into cytoplasm by a RanGTP-dependent dsRNA-binding protein, exportin 5 [9].

In the cytoplasm, another RNase III enzyme, Dicer, processes the pre-miRNAs into the miRNA:miRNA duplex of ~22 nucleotides. Generally, one chain of the miRNA duplex will bind to a multiprotein complex, named RNA-induced silencing complex (RISC). The single stranded miRNA in RISC acts as a template that recognizes the complementary mRNA, and then negatively regulates mRNA expression either by direct mRNA degradation or by translational repression, depending on the complementarity of miRNA and the target mRNA [10,11,12].

A single miRNA may have numerous target mRNAs, whereas several miRNAs may bind and regulate the same target. Therefore, miRNAs are involved in multiple biological processes, including gene regulation, apoptosis, hematopoietic development, and maintenance of cell differentiation. It is estimated that one-third of human genes are regulated by miRNAs [13]. In short, miRNAs play a key role in genomic and epigenomic interaction [14].

## 2. The Role of miRNA in Lung Cancer Development and Behavior

The development and behavior of cancer are complex. In 2000, Hanahan et al. comprehensively illustrated six hallmark capabilities of cancer, including sustaining proliferative signaling, evading growth suppressors, activating invasion and metastasis, enabling replicative immortality, inducing angiogenesis, and resisting cell death. A decade later, Hanahan and coworkers extended the original six hallmarks to eight, by adding avoiding immune destruction and dysregulating cellular energetics. Additionally, promoting inflammation with genome instability and mutation are thought to be two enabling characteristics of tumors by Hanahan et al. [15]. For each process in cancer biology, interaction between genetic preconditions and epigenetic alterations are equally important. Here, we will focus on the miRNAs involved in each hallmark capability. A brief summary of miRNAs and their associated pathways are outlined in Figure 1.

### 2.1. Sustaining Proliferative Signaling

Sustained cell proliferation and unsuppressed cell growth are thought to be the fundamental characteristics of cancer. Several genes and proteins are involved in this process, especially, some kinases and kinase receptors [16]. By far, the epidermal growth factor receptor (EGFR)-signaling pathway is the best-known example in lung cancer. Ligands such as epidermal growth factor (EGF) or transforming growth factor-α (TGF-α) bind to EGFR, resulting in its activation and receptor transphosphorylation. This process elicits a further two major signaling pathways, Ras/Raf/MEK/ERK and PI3K/Akt/mTOR, which further enhance cell proliferation and cell cycle progression [17]. miRNAs that directly target *EGFR* are miR-7, miR-27a-3p, miR-30, miR-34, miR-128, miR-133, miR-134, miR-145, miR-146, miR-149, miR-218, and miR-542-5p [16,18,19,20].

Echinoderm microtubule-associated protein-like 4 (EML4) and anaplastic lymphoma kinase (ALK) fusion proteins, which also initiate Ras/Raf/MEK/ERK and PI3K/Akt/mTOR pathways, are emerging therapeutic targets in NSCLC [21]. miR-96 has been reported as a post-transcriptional suppressor of *ALK* in cell model by Vishwamitra et al. [22].

*ROS* proto-oncogene 1 receptor tyrosine kinase (*ROS1*) is another actionable driver mutation gene in lung cancer, with available targeting therapeutics [23]. Similar to other receptor tyrosine kinases, ROS1 promotes cell survival and proliferation through activating downstream signaling pathways, such as SHP-1/SHP-2, JAK/STAT, PI3K/Akt/mTOR, and MEK/ERK [24]. miR-760 was found to reduce cell proliferation by suppressing ROS1 expression in NSCLC cell lines [25].

Kirsten rat sarcoma 2 viral oncogene homolog (KRAS), a member of the Ras protein family, is a common downstream reactor of the previously mentioned receptor tyrosine kinases, e.g., EGFR, ALK, and ROS [26]. KRAS further activates the Ras/Raf/MEK/ERK pathway, and it is estimated that 15% of lung cancer specimens harbor *KRAS* mutations [27]. The *let-7* family was able to repress KRAS and slow the proliferation of lung cancer cells both in vitro and in mouse models [28,29]. Seviour and co-workers reported that miR-193a-3p directly targets KRAS and inhibits KRAS-mutated lung tumor growth in vivo [30], while miR-181a-5p was also observed to inhibit the proliferation and migration of A549 cells through downregulating KRAS [31]. Additionally, miR-148a-3p was shown to inhibit NSCLC cell proliferation in vitro via the suppression of SOS2, a molecule upstream of the Ras signaling pathway [32]. Jiang et al. also reported that miR-1258 might target *GRB2* and inhibit Grb2, the upstream protein required for downstream Ras activation, which further downregulates the MEK/ERK pathway in mice models [33]. miR-520a-3p was reported to be involved in another downstream PI3K/Akt/mTOR pathway in vitro [34].

### 2.2. Evading Growth Suppressors

Two prototypical tumor suppressors, retinoblastoma (RB) and p53 proteins, play key roles in switching oncogenic signals off and then causing cell cycle arrest, leading to cell senescence or apoptosis. The deregulation of these pathways enables tumor cells to evade growth inhibition [15]. For example, miR-15a/miR-16 is often downregulated in NSCLC. High expression level of miR-15a/miR-16 will suppress cyclin D1, an upstream regulator of the RB pathway, causing upregulation of RB and cell cycle arrest [35]. miR-449a, targeting *E2F3*, has low expression in lung cancer tissues. The upregulation of miR-449a will cause E2F3 suppression and lead to cell cycle arrest and senescence [36]. miR-641 and miR-660 can enhance apoptosis of lung cancer cells by targeting MDM2, the upstream suppressor of the p53 pathway [37,38].

### 2.3. Enabling Replicative Immortality

Cancer cells have unlimited replicative potential (immortal) in contrast to normal (mortal) cells. The key factors are telomeres and its associated enzyme, telomerases. The telomerase is a DNA polymerase with the ability to add telomere repeat segments to the ends of telomeric DNA, causing cells to have successive generations. However, the telomerase is absent in almost all non-immortalized cells with the unavoidable fate of cell senescence [15]. The telomerase reverse transcriptase in humans (hTERT), a catalytic subunit of telomerase, is promoted by various transcription factors, such as Myc, HIF-1, and ER in different cancer cell models [39]. Additionally, many miRNAs directly target *hTERT* to control the telomere lengths. For example, miR-512-5p targets *hTERT* and suppresses tumor replication in head and neck squamous cell carcinoma, and miR-498 works in a similar way in ovarian cancers [40,41]. To date, no miRNAs have been reported to target *hTERT* in lung cancer cell studies. Apart from hTERT, telomere length is also controlled by DNA methyltransferases (DNMT). The miR-29 family exhibits a tumor-suppressive effect by targeting *DNMT3A* and *DNMT3B* in lung cancer cells [42].

### 2.4. Activating Invasion and Metastasis

The invasion–metastasis cascade, a distinct behavior of cancer, comprises a multistep process, including local invasion, intravasation, transition, extravasation, and colonization. Many miRNAs mentioned in other parts of this review are involved.

Epithelial-to-mesenchymal transition (EMT), a process central to cancer metastasis, is normally seen in the embryonic development stage, but is rarely observed in adult human cells. EMT is characterized by the loss of E-cadherin-mediated cell adhesion and the increase of cell motility, facilitating tumor invasion and metastasis [43]. The miR-200 family is the best-known example to be involved in this process. miR-200 targets zinc finger E-box-binding homeobox *(ZEB)1* and *ZEB2*, which code for the transcriptional repressors of E-cadherin. Thus, the upregulation of miR-200 leads to increased E-cadherin expression and the reduced motility of lung cancer cells [44].

### 2.5. Inducing Angiogenesis

Tumor cells need a supply of nutrients and oxygen as well as needing to eliminate waste and carbon dioxide. Neovascularization, therefore, is required for cancer development. Vascular endothelial growth factors (VEGFs) are well-known participants in angiogenesis induction, whereas thrombospodin-1 (TSP-1) is involved in angiogenesis inhibition [15]. Numerous miRNAs have been reported to regulate angiogenesis in various cancer cell lines [45]; for example, miR-107 is involved in the inhibition of angiogenesis in colon cancer cell lines by the downregulation of VEGF, via targeting the hypoxia-inducible factor-1β (*HIF-1β*) [46]. In lung cancer cell lines, members of the miR-200 family were reported to inhibit angiogenesis by targeting *VEGF* [47,48]. The upregulation of miR-126 and miR-128, which directly target *VEGF-A* and *VEGF-C*, respectively, shows the ability to inhibit angiogenesis in lung cancer cell lines [49,50]. The overexpression of miR-497 inhibits angiogenesis in NSCLC by targeting a hepatoma-derived growth factor (*HDGF*), inducing a VEGF-independent angiogenetic pathway [51]. On the other hand, miR-494 promotes angiogenesis by targeting phosphatase and tensin homolog (PTEN), a VEGF suppressor [52]. Besides, the expression of miR-23a in lung cancer-derived exosomes under hypoxic conditions enhances angiogenesis by targeting prolyl hydroxylase 1 and 2 (*PHD1* and *2*), inducing HIF-1 α and, consequently, activating the VEGF pathway. This also reflects the theory that tumor cells in a hypoxic status will develop neovascularization for their own needs [53].

### 2.6. Resisting Cell Death (Apoptosis)

Another cancer hallmark is the evasion from programmed cell death, or so-called apoptosis. There are two major pathways leading to cell apoptosis: Extrinsic and intrinsic pathways. The extrinsic pathway is initiated by receiving extracellular death-inducing signals via cell surface receptors, such as the Fas receptor, whereas the intrinsic pathway starts with various genotoxic agents, metabolic insults, or transcriptional cues sensed by BH3-only proteins, and results in the inactivation of some BCL-2 family members. Both pathways eventually lead to the activation of caspases, which induce further cell destruction [15]. Wang et al. reported that miR-16-1 downregulates Bcl-2 and induces cell apoptosis in A549 cells [54]. Tian et al. showed that miR-130b suppressed lung cancer cell apoptosis by indirectly upregulating Bcl-2 via the peroxisome proliferator-activated receptor gamma (PPAR-γ)/VEGF pathway [55]. This is also an example demonstrating the crosstalk between angiogenetic and apoptotic pathways.

### 2.7. Deregulating Cellular Energetics

To meet high biosynthetic and bioenergetic demands for uncontrolled proliferation, cancer cells develop different energy metabolism modes. The Warburg effect is the best-described model, which demonstrates the reprogrammed metabolic pathway in cancer cells, using the so-called “aerobic glycolysis” [15]. For example, the downregulation of miR-144 in lung cancer cells may upregulate the glucose transporter (GLUT1) expression and increase glucose uptake [56]. miR-33b negatively regulates lactate dehydrogenase A (LDHA), an enzyme needed in glucose metabolism, and inhibits NSCLC cell growth [57]. HIF-1 is known to enhance the expression of many glycolytic enzymes, and miR-199a has been reported to inhibit lung cancer proliferation by suppressing HIF-1α and affecting the glycolytic pathway [58]. The inhibitor of HIF-1α (FIH) is downregulated by miR-31-5p; thus, miR-31-5p promotes lung cancer progression via enhancing glycolysis [59].

### 2.8. Avoiding Immune Destruction and Tumors Promoting Inflammation

The interaction between cancer cells and the immune system is complex. There are two known phenomena. First, tumor cells can evade immune surveillance and avoid subsequent immune-induced cell destruction. Second, the immune cells infiltrating into the tumor microenvironment lose the ability to eradicate cancer cells and instead, paradoxically, supply the tumor cells with the growth factors, survival factors, and proangiogenic factors, supporting their progress and survival. The former process is achieved by the programmed death-ligand 1 (PD-L1)/PD-1 pathway. PD-1 is located on the surface of T-cells, whereas PD-L1 is found on tumor cells. When PD-1 interacts with PD-L1, inhibitory signals are initiated and prevent subsequent tumor cell eradication [16].

miRNAs are also involved in this process. The miR-34 family, especially miR-34a, is associated with PD-L1 expression. miR-34a may bind to the 3′UTR of *PD-L1* mRNA and suppress PD-L1 expression [60]. miR-197, another microRNA involved in PD-L1 regulation, is negatively associated with PD-L1 expression and overall survival in NSCLC patients. The regulation of PD-L1 by miR-197 was enhanced indirectly by CKS1B (CDC28 protein kinase regulatory subunit 1B) and STAT3 (signal transducer and activator of transcription 3), and the latter would bind to the promoter region of the *PD-L1* gene [61]. The miR-200 family, including miR-200a, miR-200b, miR-429, miR-200c, and miR-141, are reported to target *PD-L1* expression and target *ZEB1* in NSCLC. The lower the miR-200 expression, the more the suppression of CD8+ cell infiltration observed as a consequence of PD-L1 upregulation. And this may enhance tumor cell proliferation and metastases [44].

### 2.9. Genome Instability and Mutation

Genomic instability, which means the increased tendency of genome alteration during cell division, is an important initiative feature of cancer. This phenomenon is actually present in all types and stages of human cancers. On the contrary, normal cells preserve their genomic integrity by using multiple surveillance mechanisms, e.g., DNA damage checkpoints, DNA repair machinery, and mitotic checkpoints [62]. Any impairment of these mechanisms confers genomic instability.

miRNAs play an important role in maintaining integrity by regulating cellular DNA sensing and repair mechanisms. Firstly, miRNA genes are frequently located in fragile chromosomal sites, indicating that miRNA disruption could be one of the early events leading to genomic instability and thereby initiating carcinogenesis [63]. For example, the 11q23-q24 genomic region that is frequently deleted in lung cancers contains the gene for miR-34a, a well-known tumor suppressor miRNA [64]. Secondly, miRNAs may dysregulate cell cycle checkpoints, such as p21 and p27, causing early or delayed entry to the next phase. This leads to abnormal genomic contents, immature chromosome separation, and consequently causes genomic instability and cancer. Thirdly, DNA damage responses, including the detection and repair of damaged DNAs, are crucial for maintaining genomic stability. Zhang et al. reported that miR-128-3p accelerates chromosomal instability and cell cycle arrest by suppressing spectrin alpha, non-erythrocytic 1 (SPTAN1) [65]. Another important gate-keeper, p53, acts as a sensor in DNA damage response that functions to block the cell cycle to repair damaged DNA or drive cells to apoptosis if there is no potential for repair. Several miRNAs modulate p53 expression by direct targeting; for instance, miR-125b and miR-504. miR-29 has been reported to indirectly upregulate p53 by targeting *PIK3R1* and *CDC42* and inducing apoptosis [63].

## 3. The Interaction of Cigarette Smoke (CS) and miRNAs in Lung Cancer Tumorigenesis

Long-term exposure to CS, which contains more than 70 carcinogens, is a well-established risk factor of lung cancer [66,67]. With increasing awareness in the human population of the potential harm, cigarette consumption is decreasing worldwide. However, the incidence of lung cancer has not declined accordingly. Air pollution and particulate matter are other important risk factors of lung cancer.

CS is linked to lung cancer formation via genomic alteration and epigenomic aberrant expression [68,69]. Genetically, carcinogens in CS might induce mutations in critical growth regulatory genes, such as *KRAS* and *TP53* [68].

In addition to genomic alteration, exposure to CS also induces epigenomic dysfunction. Firstly, smokers have differential DNA methylation profiles, which might interfere with the normal genetic expression in the small airway epithelium [69]. Secondly, CS induces histone modifications [70]. Sundar et al. reported that CS causes site-specific post-translational histone modifications (PTMs) of histones H3 and H4, which might play a role in lung cancer development [71]. Thirdly, CS might alter miRNA expression. Many deregulated miRNAs are found to be associated with tumor suppressor gene-silencing or oncogene upregulation. Advani et al. reported examples of dysregulated miRNAs using microarray-based approaches and bioinformatic analyses. In their H292 cell line with CS exposure, the downregulation of miR-155-5p that targets CCAAT/enhancer binding protein—beta *(CEBPB)*, and upregulation of miR-21-5p that targets *PDCD4*, were observed, respectively [72]. In a recent study, a sound mechanism was proposed using human bronchial epithelium cells that posited CS might induce HIF-1α-dependent miR-21 upregulation, reducing PTEN levels and then activating the Akt/NF-κB pathway, leading to malignant transformation [73]. Nicotine, one of the various constituents in CS, is not thought to be a mutagen. However, nicotine is reported to promote proliferation and metastasis by suppressing miR-99b and miR-192 and upregulating fibroblast growth factor receptor 3 (FGFR3) and RB1 levels respectively, in NSCLC cell lines [74].

## 4. MicroRNA and Tumor Microenvironment

Over the past several decades, the complexity of the interactions between tumors and their surrounding environments has attracted much attention [75,76]. Viewed from this perspective, tumor biology could be understood by studying the surrounding specialized cell types, as well as their constructed “tumor microenvironment” (TME) during tumorigenesis. The TME compromises immunosuppressive cells, such as tumor-associated macrophages (TAMs), cancer-associated fibroblasts (CAFs), regulatory T-cells (Tregs), myeloid-derived suppressor cells (MDSCs) and others that are involved in the TME and carcinogenesis. miRNAs regulate different stromal cells and exert their impact on the regulation of interactions between cancer cells and their surrounding stromal cells [75].

### 4.1. Tumor-Associated Macrophages

TAMs, also known as alternatively activated macrophages, can foster local invasion at the tumor periphery by supplying matrix-degrading enzymes, such as metalloproteinases and cysteine cathepsin proteases. They directly promote non-resolving inflammation, which leads to proliferation, invasion, metastasis, tumor angiogenesis, inhibition of the immune response mediated by T-cells, and the promotion of tumor progression. miR-320a has been known to promote M2-like phenotype through the downregulation of STAT4 [77]. Expression of miR-130a is at a high level in M1 macrophages. When knocking down miR-130a, macrophages skew from M1 to M2 phenotypes [78].

### 4.2. Tumor-Associated Fibroblasts

CAFs consist of at least two distinct cell types: One similar to the fibroblasts supporting most normal epithelial tissues, and the other comprised of myofibroblasts that differ from those of tissue-derived fibroblasts. The presence of myofibroblasts and fibroblastic cells lead to enhanced tumor phenotypes, proliferation, angiogenesis, and invasion and metastasis. The secretion of miR-21 in CAFs has been reported to support lung cancer progression [79]. The high expression of miR-101 in CAFs suppresses CXCL12 expression, which impairs CAFs from enhancing tumor cell proliferation and sphere formation. The high levels of miR-101 lead to tumor cell apoptosis [80]. High levels of stromal cell-derived factor 1 (SDF-1) in CAFs, regulated by low levels of miR-1, enhance lung cancer cell A549 proliferation and drug resistance [81].

### 4.3. Tumor Endothelial Cells and Tumor-Associated Angiogenesis

The development, differentiation, and homeostasis of endothelial cells are critical for tumor-associated vasculature for blood and nutrient supply. Tumor endothelial cells, which line tumor blood vessels, exhibit altered phenotypes compared with normal counterparts. They demonstrate activated proliferation and migration and upregulate several angiogenesis-related genes [82]. Some endothelial-specific miRNAs have been involved in various aspects of angiogenesis as proliferation, migration, and morphogenesis of endothelial cells [83]. In one study, several miRNAs including miR-142-3p, miR-143-3p, etc. derived from inhibited lung cancer extracellular vesicles. Our previous study showed that hypoxia-induced exosomal miR-23a downregulates prolyl hydroxylase 1/2 (PHD 1/2) and accumulates hypoxia-inducible factor-1α (HIF-1α) in the endothelia [53].

An angiogenic switch activates quiescent endothelial cells to form new blood vessels in addition to VEGF, angiopoietin, and FGF signals. Another possible pathway to study is the surface markers or gene profiles between normal and tumor endothelia. miR-494, in lung cancer, targets *PTEN*, and the subsequently activated Akt pathway promotes angiogenesis [52]. One study suggested that angiogenesis-related miRs were altered, including miR-21, miR-106a, miR-126, miR-155, miR-182, miR-210, and miR-424, but these observations have not been validated [84].

### 4.4. The Epithelial-To-Mesenchymal Transition

The extracellular matrix (ECM) participates in steps involved in cancer metastasis. Many ECM proteins are dysregulated during the progression of cancer, both biochemically and biomechanically. Many ECM proteins are associated with the induction of epithelial-to-mesenchymal transition (EMT) [85,86]. EMT enables epithelial cancer cells to invade and metastasize [86]. There are several microRNAs regulating EMT-related genes. miR-149 targets Forkhead box M1 (*FOXM1*) to suppress EMT in A549 [87]. miR-33a targets Twist1 and inhibits EMT-related metastasis in NSCLC. Low levels of miR-33a carries worse prognosis [88].

Taken together, miRs play roles in TME by controlling cancer cellular behaviors and dysregulating their interaction with surrounding cells. Investigating the relationships between cancer cells and their microenvironment is necessary in lung cancer diagnosis and treatment.

## 5. The Role of miRNAs in Lung Cancer Diagnosis

### 5.1. Histologic Subtype Identification

miRNAs could be used to differentiate subtypes among lung cancers. For example, miR-205 is reported to be specific to squamous cell carcinoma (SqCC) among NSCLC samples [89,90], whereas miR-124a is specific to adenocarcinoma [91]. In another report, Zhang et al. found four high expressions of miRNAs (miR-205, miR-93, miR-221, and miR-30e) in SqCC, and five highly expressed miRNAs (miR-29b, miR-29c, let-7e, miR-100, and miR-125a-5p) in adenocarcinoma [92]. As for SCLC, miR-375 and miR-21-5p are reported to be highly expressed levels [93,94].

The miRNA expression also interferes with growth patterns of lung adenocarcinoma. Nadal et al. analyzed the histological growth patterns among lung adenocarcinomas showing that high expression levels of miR-27a, miR-212, and miR-132 were significantly associated with the presence of solid components in the tumor, whereas miR-30d was associated with lepidic patterns or mucinous invasive adenocarcinoma [95].

“miRview lung” distinguished four main lung cancer subtypes by evaluating the expression levels of eight miRNAs, including miR-106a, miR-125a-5p, miR-129-3p, miR-205, miR-21, miR-29b, miR-375, and miR-7. This assay was validated independently in 451 other pathologic or cytologic samples, with an overall diagnostic accuracy of 94% [96].

Furthermore, Rosenwald et al. validated 204 independent samples (nearly 50% were metastatic tumors) through 48 miRNAs with a success rate of 85% [97].

### 5.2. Distinguishing Primary from Metastatic Tumors

Distinguishing primary lung tumors from metastatic ones from other sites would be a critical issue for clinicians. Barshack et al. reported that the high expression of miR-182 was associated with primary lung tumors, whereas a high expression of miR-126 was detected in metastatic tumors [98]. Highly expressed levels of miR-552 and miR-592 have been used to differentiate metastatic colorectal adenocarcinoma from primary lung adenocarcinoma [99].

### 5.3. Lung Cancer Screen

Molecular biomarkers have been an adjunctive tool for lung cancer screening. Among them, circulating miRNAs demonstrated a fascinating potential for diagnostic benefit. The reason is based on feasibility in detecting miRNAs from body fluids, such as blood, colostrum, urine, and pleural fluid. The second advantage is that circulating miRNAs are very stable in body fluid, without effects of endogenous RNases. Thirdly, circulating miRNAs collect all pathologic signals from different tumor parts or metastatic sites, overcoming the problem of tumor heterogeneity [100].

Currently, numerous miRNAs have been used as a diagnostic tool for lung cancer. Many of them are integrated as a “set” or “kit” to detect the possibility of lung cancer. For example, miR-test is a serum-based miRNA test that measures the signature of 13 miRNAs, including miR-92a-3p, miR30b-5p, miR-191-5p, miR-484, miR-328-3p, miR-30c-5p, miR-374a-5p, let-7d-5p, miR-331-3p, miR-29a-3p, miR148a-3p, miR-223-3p, and miR-140-5p [101]. In high-risk lung cancer patients, the miR-test exhibits diagnostic accuracy, sensitivity, and specificity of 74.9%, 77.8%, and 74.8%, respectively. Another commercial kit, microRNA signature classifier (MSC), used the expression ratio of 24 predefined miRNAs in lung cancer with reported sensitivity, specificity, positive predictive value, negative predictive value, and positive likelihood ratio of 87%, 81%, 27%, 98%, and 4.67%, respectively [102].

## 6. The Role of miRNAs in Lung Cancer Prognosis

OncomiRs and tumor suppressor miRNAs as oncogenes and tumor suppressor genes are playing major roles in lung cancer prognosis. In 2004, the first study by Takamizawa et al. revealed that lower expressions of *let-7* microRNA was significantly associated with shorter survival in patients with surgical lung cancer [103]. In 2014, a meta-analysis revealed low expression of *let-7* microRNA as a poor prognosis predictive factor in lung cancer patients by Xia et al. [104]. miR-155 is associated with poorer recurrence-free survival in NSCLC [105].

## 7. The Role of miRNAs in Lung Cancer Therapy

miRNAs modify the response to chemotherapy, radiotherapy, and targeted therapy [106]. A brief summary and associated miRNAs are listed in Table 1.

### 7.1. Chemotherapy

Platinum-based therapy is the most common regimen for treating lung cancer, both for SCLC or NSCLC. miRNAs are related to chemotherapy sensitivity or resistance. Enhanced miR-106b expression suppresses the polycystin 2 (PKD2) level, and subsequently, leads to the downregulation of P-glycoprotein, which mediates the efflux of multiple anticancer drugs; consequently, cisplatin sensitivity was increased in an NSCLC cell line with miR-106b upregulation [107]. miR-503 inhibited the drug efflux mechanism and enhanced cisplatin sensitivity by suppressing several drug resistance-associated proteins, e.g., MDR1, MRP1, ERCC1, survivin, and Bcl-2 [108]. On the contrary, miR-196 upregulation was associated with increased MDR1, MRP1, ERCC1, survivin, and Bcl-2, inducing drug efflux and cisplatin resistance [109]. Both miR-15b and miR-27a conferred resistance to cisplatin therapy by targeting *PEBP4*- and *RKIP*-mediated EMT [110,111].

**Table 1 ijms-20-01611-t001:** miRNAs associated with sensitivity or resistance to lung cancer therapies.

Chemo-sensitive			Chemo-resistant		
miRNA	Target/pathway	Ref.	miRNA	Target/pathway	Ref.
*let-7*	LIN28	[112]	miR-15b	PEBP4	[110]
miR-7	EGFR	[113]	miR-21	PTEN, SMAD7	[114,115]
miR-17, 20a, 20b	TGFβR2	[116]			
miR-17 and miR-92 family	CDKN1A, RAD21	[117]	miR-27a	RKIP	[111]
miR-34a	PEBP4	[118]	miR-92a	PTEN	[119]
miR-101	ROCK2	[120]	miR-106a	?	[121]
miR-106b	PKD2	[107]			
miR-135b	FZD1	[122]	miR-181c	WIF1, Wnt/β-catenin pathway	[123]
			miR-196	up-regulation of MDR1, MRP1, ERCC1, Survivin and Bcl-2	[109]
miR-137	NUCKS1	[124]	miR-205	PTEN, Mcl-1 and Survivin	[125,126]
miR-138	ZEB2	[127]	miR-488	eIF3a-mediated NER signaling pathway	[128]
miR-146a	CCNJ	[129]			
miR-181b	TGFβR1/Smad signaling pathway	[130]			
miR-184	Bcl-2	[131]			
miR-218	RUNX2	[132]			
mir-296	CX3CR1	[133]			
miR-379	EIF4G2	[134]			
miR-451	c-Myc-survivin/rad-51 signaling	[135]			
miR-503	Down-regulation of MDR1, MRP1, ERCC1, Survivin and Bcl-2	[108]			
miR-9600	STAT3	[136]			
**Radio-sensitive**			**Radio-resistant**		
miRNA	Target/pathway	Ref.	MiRNA	Target/pathway	Ref.
miR-29c	Bcl-2, Mcl-1	[137]	miR-21	PTEN, HIF1α, PDCD4	[114,138]
miR-138	SENP1	[139]	miR-25	BTG2	[140]
miR-200c	PRDX2, GAPB/Nrf2, and SESN1	[141]	miR-210	HIF1	[142]
miR-328	γ-H2AX	[143]	miR-1323	PRKDC	[144]
miR-449a	?	[145]			
miR-451	c-Myc-survivin/rad-51 signaling	[135]			
**EGFR TKI-sensitive**			**EGFR TKI-resistant**		
miRNA	Target/pathway	Ref.	miRNA	Target/pathway	Ref.
miR-126	Akt and ERK pathways	[146]	miR-21	PTEN, PDCD4	[147]
miR-134/miR-487b/miR-655 cluster	MAGI2	[148]	miR-23a	PTEN/PI3K/Akt pathway	[149]
miR-200c	PI3K/Akt pathway	[150]	miR-30 family	PI3K-SIAH2	[151]
miR-223	IGF1R/PI3K/Akt pathway	[152]	miR-214	PTEN/AKT pathway	[153]
miR-483-3p	integrin β3/ FAK/Erk pathway	[154]			
**ALK TKI-sensitive**			**ALK TKI-resistant**		
miRNA	Target/pathway	Ref.	miRNA	Target/pathway	Ref.
miR-200c	ZEB1	[155]			

### 7.2. Radiation Therapy

miRNAs regulate cell cycles by altering cell proliferation and apoptosis, interfering with damaged DNA repair, and enhancing tumor angiogenesis. Clinical applications of miRNAs in lung cancer radiotherapy have been reported. Taking miR-200c as an example, it can enhance the effect of radiotherapy on lung cancer cells by repressing oxidative response genes and inhibiting DNA repair [141]. Effects for other similar miRNAs such as miR-449a [145], miR-138 [139], miR-25 [140], and *let-7* [156] have been reported.

By using bioinformatical analysis, Chen et al. identified four plasma miRNAs including miR-98-5p, miR-302e, miR-495-3p, and miR-613 as a predictor of a radiotherapy response in NSCLC [157].

### 7.3. Molecular Targeted Therapy

#### 7.3.1. EGFR-Tyrosine Kinase Inhibitors (TKI)

A number of studies have reported a linkage between the altered expression of a variety of miRNAs and sensitivity to gefitinib and erlotinib [158]. For example, miR-21 induced gefitinib resistance by activating ALK and ERK, and suppressing PTEN in NSCLC [159,160]. Bisagni et al. reported that upregulation of miR-133b was associated with longer progression-free survival when erlotinib was used as second- or third-line therapy for NSCLC. [161]. To add a miR-34a mimic on erlotinib showed strong synergistic primary, or even, acquired resistance to EGFR-TKI in NSCLC cells [162]. The synergistic effect was also present with newer generation TKIs, such as afatinib and osimertinib [163]. The expressions of miR-1, miR-124, and miR-196a were found to be high in T790M-mutated NSCLC, however, the role of these miRNAs was still unclear [164].

#### 7.3.2. ALK-TKI

*ALK* fusion-positive lung cancer comprises a small percentage of the NSCLC population with currently available therapeutic tyrosine kinase inhibitors, such as crizotinib [165,166,167]. A high expression of miR-1343-3p and low expression of miR-6713p, miR-103a-3p, let-7e and miR-342-3p were significantly distinctive in the *ALK*-positive group, compared to the *EGFR*-positive and *KRAS*-positive groups [166]. A panel of plasma miR-28-5p, miR-362-5p and miR-660-5p expression was used to predict *ALK*-positive NSCLC, with the areas under the receiver operating characteristic curves of 0.876. Besides this, the pre-treatment miR-362-5p level was positively associated with progression-free survival, and post-treatment miR-660-5p level predicted a better therapeutic response of crizotinib, respectively [167].

miRNAs play a role mediating ALK inhibitor resistance. Yun et al. reported that *ALK*-positive lung cancers acquire resistance to ALK inhibitors through histone H3 lysine 27 acetylation (H3K27ac) loss and the repression of miR-34a, leading to *AXL* activation [168]. Another study demonstrated that the high expression of miR-200c reversed EMT by direct targeting of *ZEB1* and indirect upregulation of E-cadherin by using a crizotinib-resistant lung cancer cell line (NCI-2228/CRI) [155].

### 7.4. Immunotherapy

Halvorsen et al. found a signature of seven circulating miRNAs, miR-215-5p, miR-411-3p, miR-493-5p, miR-494-3p, miR-495-3p, miR-548j-5p, and miR-93-3p, significantly associated with increased overall survival in NSCLC patients using nivolumab [169].

A liposomal miR-34a mimic, MRX34, was tested in a mouse model with lung cancer, and showed significant downregulation of PD-L1. The simultaneous prescription of MRX34 with radiotherapy would lead to a greater increase in CD8+ T-cells and decrease in PD1+ T-cells. The authors concluded that the combination of miRNA therapy with standard therapy might be a novel approach for cancer treatment [60].

### 7.5. miRNA-Targeted Therapeutics Using miRNA Mimics or Anti-miRNAs

Based on the understanding of interactions between miRNAs and tumor genesis and behavior, therapies targeting miRNAs are proposed. Both miRNA mimics and anti-miRNAs have been successfully applied to restore balanced gene networks in lung cancer cell lines and xenograft animal models.

To achieve gene network balance, two approaches may be employed: One is to replace downregulated miRNAs by using synthetic miRNA mimics or miRNA-expressing vectors; the other is to reduce upregulated miRNAs by, most commonly, adding chemically modified antisense nucleotides (anti-miRNAs).

In the field of lung cancer, miR-150, *let-7*, miR-34, miR-29b, miR-200c, and miR-145 are the most studied therapeutic miRNAs. The results from a large number of preclinical trials appear promising.

mir-34a targeted *PD-L1*, altering the p53/mir-34/PD-L1 pathway [60]. MRX34, a liposomal miR-34a mimic, was the first miRNA therapy introduced into phase I clinical trials, and has shown anticancer activity in solid tumors [170]. However, the phase I clinical trial was halted prematurely due to serious immune-related adverse effects [16].

The MesomiR 1 trial, another phase I trial for miRNA therapy applied to thoracic cancer treatment, is ongoing (ClinicalTrials.gov Identifier: NCT02369198). Patients with malignant pleural mesothelioma or advanced NSCLC, treated with previous systemic therapies, were enrolled to evaluate the optimal dose of TargomiR, a nano-cell treatment packed with synthetic miR-16 [171].

## 8. Current Limitations and Future Perspectives

Using cell-free miRNA to diagnose lung cancer or to monitor the treatment response is attractive; however, the major obstacle to the application of miRNA for diagnostic purposes is the lack of reproducibility between different studies. Reasons for such inconsistencies may be, for instance, the inter-individual variability in the serum miRNAs level. Besides, few large-scale investigations have been undertaken, and most studies presently underway are underpowered due to limited case numbers. Moreover, variation of technical aspects between different studies, e.g., RNA purification procedures, measurement platforms, statistical methodologies, etc., are all attributed to the discordance [172].

As for therapeutic purposes against lung cancer, we wish to highlight some issues. Firstly, miRNA-targeted therapies should be highly tissue-specific when delivering miRNA mimics or inhibitors. Secondly, miRNA mimics or inhibitors should be efficiently taken up by cancer cells with their effects maintained for a certain peroid. Thirdly, unwanted off-target effects and toxicities should be minimal. Currently, the delivery system of miRNAs can be roughly divided into viral- or non-viral-based systems. Retroviruses, adenoviruses, and adeno-associated viruses are representative viral vectors. They have high transfection efficiency for delivering miRNAs and these viral vectors have limited replication ability. However, there are still some concerns regarding their usage; for instance, toxin production, immunogenicity that induces inflammatory responses, and mutations caused by the inserted sequence [173]; therefore, viral-based delivery of miRNAs is less favored clinically [174].

Non-viral systems such as lipoplex or polymer-based methods demonstrate lower toxicity and lower immunogenicity. Although they are much safer, their main disadvantage is the insufficient miRNA delivery rate. Among these, the most promising methods are locked nucleic acid (LNA) and liposomal nanoparticles of miRNA mimics. They both change the pathway regarding rapid degradation of miRNAs in biofluid and increase the delivery efficiency in vivo [174]. A phase II clinical trial using LNA-anti-miR (miravirsen or SPC3649; Santaris Pharma, Denmark) to target miR-122, for treating chronic hepatitis C virus infection, was completed with good tolerance. In addition, LNA-anti-miR showed significant antiviral activity [175,176].

Due to the non-specific targets of miRNAs, unwanted off-target effects ensue if the delivered miRNA mimics, or produces an anti-miRNAs effect, targeting unintended mRNAs. Besides this, the exogenous miRNA will compete with miRISC and endogenous miRNAs, interfering with normal regulation [177]. Beyond the off-target effects, an alteration of immune response is observed. For example, a grade 4 cytokine release syndrome was observed in a participant in the phase I clinical trial of MRX34, causing premature closure of this trial [16]. Thus, it is important to develop the delivery tool with safety and efficiency in the future.

## 9. Conclusions

Numerous studies focusing on miRNAs have been published. Much effort has been made to adapt the results into clinical use, such as miR-122 inhibition by miravirsen in the treatment of hepatitis C [175]. miRNAs are demonstrated to be involved in the pathogenesis, diagnosis, and prognosis of lung cancer; however, as for lung cancer treatment, the results of clinical trials have not been encouraging. More comprehensive studies are needed to demonstrate whether miRNA targeted therapy can benefit lung cancer patients or not.

## Figures and Tables

**Figure 1 ijms-20-01611-f001:**
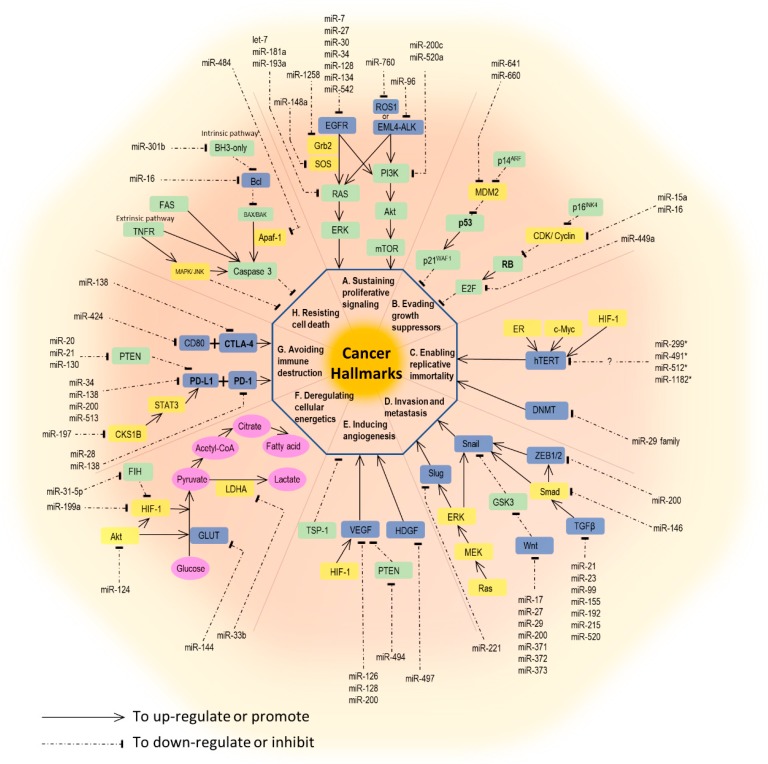
A diagram of microRNAs and their involved pathways to cancer hallmarks. A. Epidermal growth factor (EGF) and its receptors, EGFR, conduct proliferative signaling for lung cancer cells through downstream RAS/ERK and PI3K/Akt/mTOR pathways. EML4-ALK and ROS1 also initiate cancer cell proliferation through similar pathways. The involved miRNAs are illustrated. B. Deregulations of tumor suppressors, RB and p53, lead the cancer cells evade from growth inhibition. C. The telomerase reverse transcriptase in humans (hTERT) links to the immortality of cancer cells. miR-299, miR-491, miR-512 and miR-1182 are reported to target hTERT. However, these involvements are studied in various cancer cells other than lung cancer. Additionally, miR-29 family can target DNA methyltransferases (DNMT) and control the telomere lengths. D. Snail, Slug, and Wnt are key players involved in epithelial to mesenchymal transition (EMT) with implication of tumor metastasis and invasion. The associated miRNAs are shown in the diagram. E. Vascular endothelial growth factors (VEGF) stand for key participants in promoting tumor angiogenesis. miR-126, miR128 and miR-200 may target VEGF. F. Cancer cells develop aerobic glycolysis as the reprogrammed metabolic pathway. For example, downregulation of miR-144 is found in lung cancer cells with upregulated glucose transporter (GLUT1) expression and increased glucose uptake. G. The interaction of programmed death-ligand 1 (PD-L1) and its receptor (PD-1) leads the cancer cells evading from immune destruction. miR-34, miR-138, miR-200 and miR-513 target PD-L1 and suppress its expression. H. Fas receptors (intrinsic pathway) and BH3-only proteins (extrinsic pathway) are key participants in the resistance to cell apoptosis. miR-301b targeting BH3-only proteins, and miR-16 targeting Bcl are reported to be involved in cell apoptosis. Note: This diagram is simplified where major, but not all, pathways are illustrated. Meanwhile, cross-talk and interactions between different pathways, not shown in this diagram, might exist. For example, the EGFR signaling pathway might not only promote cell proliferation but also enhance invasion, metastasis, angiogenesis, and resistance to apoptosis.

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
