# Peer review of "The Roles of MicroRNA in Lung Cancer"

_ijms, 2019, doi:10.3390/ijms20071611_

Reviewer 1 Report

Wu et al have reviewed the role and significance of miRNAs in lung cancer. They have constructed their review based on the well-established hallmarks of cancer and have tried to review the publications in each category. In the second part of their review they have described the clinical utility of miRNAs in diagnosis, prognosis, and therapy.

First and foremost, this review is attempting to cover an ever expanding subject with vast number of studies and thus will have deficiencies in every field. I suggest that the authors choose a number of specific areas and while having their general review on the subject, perform a deeper and more focused review on their selected topics.

One of the main issues with the manuscript is the unacceptable number of language errors in the text. I strongly recommend that the authors send their manuscript to an English editor with adequate knowledge of the subject before returning the manuscript for consideration.

Furthermore the scientific language used to describe the relationships between the microRNAs and their targets and the mechanism of their effects are often misleading. For instance authors often use “overexpression” to refer to the associations of the higher levels of a specific miRNA in clinical samples. Since in these types of studies no overexpression can be performed, they should describe the findings as the correlation or association of high/low levels.

They should also be clear when referring to specific experimental settings and clarify if an overexpression of knockdown effect has been seen in cell culture, or animal studies. In general any cell culture study that lacks an in vivo component is not considered significant enough to be mentioned in a higher level review.

Even though arranging the subheadings based on the established hallmarks of cancer is an acceptable strategy, the authors have neglected a number of very exciting and interesting aspects of recent studies in the field. The most important of these is the role of miRNAs in tumor stroma or non-malignant components of the tumor. A number of studies have established that tumor growth is the result of the interaction between malignant cells and stromal cells, including fibroblasts, endothelial cells, and most notably immune cells. Perhaps each of the hallmarks can be divided based on the miRNAs involved in the malignant and non-malignant component.

Another critical and very exciting aspect of the recent studies is the role of miRNAs downstream of specific driver mutations or signaling cascades. Even though authors have mentioned examples in the section on molecular therapy, perhaps a more scientific strategy would be to first describe the mutations or signaling cascades being targeted, then the established role of miRNAs in that pathway, and lastly the potential use of that information in clinic.

Regarding the role of cigarette smoke in inducing lung cancer, perhaps addition of a section on the effect of cigarette smoke on lung microRNAs and their relationship with tumorigenesis is warranted.

The review also lacks a description of the basic miRNA machinery in the cell (e.g dicer and Drosha) and the studies performed on the role of these molecular machines in cancer.

Author Response

Response to the Comments from Reviewer 1

"Thanks for reviewing our article, and we also have English editing from a team of native English speakers. The English editing proof is attached"

Wu et al have reviewed the role and significance of miRNAs in lung cancer. They have constructed their review based on the well-established hallmarks of cancer and have tried to review the publications in each category. In the second part of their review they have described the clinical utility of miRNAs in diagnosis, prognosis, and therapy.

First and foremost, this review is attempting to cover an ever expanding subject with vast number of studies and thus will have deficiencies in every field. I suggest that the authors choose a number of specific areas and while having their general review on the subject, perform a deeper and more focused review on their selected topics.

We agree the reviewer’s suggestions about the deficiencies in this review article. The comments are quite important to us. Due to a wide range of lung cancer, we were trying to cover the vast number of studies in miRNAs controlling or regulating lung cancer. We used the hallmarks of cancer to limit the spectra of this review article. However, any missing part in the hallmarks of cancer will loss the idea of this article. After discussion with all the authors and other specialists in this field, we keep the hallmarks of cancer as mainstay of this article. We beg your pardon as we kept the frame of it.

 One of the main issues with the manuscript is the unacceptable number of language errors in the text. I strongly recommend that the authors send their manuscript to an English editor with adequate knowledge of the subject before returning the manuscript for consideration.

We thanks the reviewer’s comment. As a non-native English speaker, we have tried our best to write and avoid some major grammar mistakes. However, there were still some major scientific language errors. Therefore, we have sent this article for English editing by native English speaking editors in MDPI. I do believe after undergoing English language editing, it will fit the requirement for publication.

 Furthermore the scientific language used to describe the relationships between the microRNAs and their targets and the mechanism of their effects are often misleading. For instance authors often use “overexpression” to refer to the associations of the higher levels of a specific miRNA in clinical samples. Since in these types of studies no overexpression can be performed, they should describe the findings as the correlation or association of high/low levels.

As requested, we have modified the erroneous scientific language accordingly. Also, this article has been modified by a team of native English speakers by MDPI.   

1.     Line 326. In another report, Zhang et al. found four high expression of miRNAs (miR-205, miR-93, miR-221, and miR-30e) in SqCC,

2.     Line 415. The expressions of miR-1, miR-124, and miR-196a were found to be high in T790M-mutated NSCLC

3.     Line 430. Another study using a crizotinib-resistant lung cancer cell line (NCI-2228/CRI) demonstrated that the high expression of miR-200c

They should also be clear when referring to specific experimental settings and clarify if an overexpression of knockdown effect has been seen in cell culture, or animal studies. In general any cell culture study that lacks an in vivo component is not considered significant enough to be mentioned in a higher level review.

As suggested, we limited the data of cell lines study and used more human data. Such as in line 130 “a high expression level of miR-15a/miR-16 will suppress cyclin D1, an upstream regulator of the RB pathway, causing upregulation of RB and then cell cycle arrest” and line 131 “miR-449a, targeting E2F3, has low expression in lung cancer tissues”

Even though arranging the subheadings based on the established hallmarks of cancer is an acceptable strategy, the authors have neglected a number of very exciting and interesting aspects of recent studies in the field. The most important of these is the role of miRNAs in tumor stroma or non-malignant components of the tumor. A number of studies have established that tumor growth is the result of the interaction between malignant cells and stromal cells, including fibroblasts, endothelial cells, and most notably immune cells. Perhaps each of the hallmarks can be divided based on the miRNAs involved in the malignant and non-malignant component.

Thanks for your suggestions. We missed the important progression in carcinogenesis and cancer progression. As suggested, we added the important part as “MicroRNA and Tumor Microenvironment” in line 271. We mentioned about TAM, TAF, angiogenesis and EMT as bellow. 

"

4. MicroRNA and Tumor Microenvironment

Over the past several decades, the complexity of the interactions between tumors and their surrounding environments has attracted much attention [78,79]. Viewed from this perspective, tumor biology could be understood by studying the surrounding specialized cell types, as well as their constructed “tumor microenvironment, TME” during tumorigenesis. The TME compromises immunosuppressive cells, such as tumor-associated macrophages (TAMs), cancer-associated fibroblasts (CAFs), regulatory T-cells (Tregs), myeloid-derived suppressor cells (MDSCs), and others which are involved in the TME and carcinogenesis. miRNAs regulate different stromal cells and exert their impact on the regulation of interactions between cancer cells and their surrounding stromal cells [78].

4.1. Tumor-Associated Macrophages

TAMs, also known as alternatively activated macrophages, can foster local invasion at the tumor periphery by supplying matrix-degrading enzymes, such as metalloproteinases and cysteine cathepsin proteases. They directly promote non-resolving inflammation, which leads to proliferation, invasion, metastasis, tumor angiogenesis, inhibition of the immune response mediated by T-cells, and the promotion of tumor progression. miR-320a has been known to promote M2-like phenotype through the downregulation of STAT4 [80]. Expression of miR-130a is at a high level in M1 macrophages. When knocking down miR-130a, macrophages skew from M1 to M2 phenotypes [81].

4.2. Tumor-Associated Fibroblasts

CAFs consist at least two distinct cell types: one similar to the fibroblasts supporting most normal epithelial tissues, and the other comprising of myofibroblasts which differ from those of tissue-derived fibroblasts. The presence of myofibroblasts and fibroblastic cells lead to enhanced tumor phenotypes, proliferation, angiogenesis, and invasion and metastasis. The secretion of miR-21 in CAFs has been reported to support lung cancer progression [82]. The high expression of miR-101 in CAFs suppresses CXCL12 expression, which impairs CAFs from enhancing tumor cell proliferation and sphere formation. The high levels of miR-101 lead to tumor cells apoptosis [83]. High levels of stromal cell-derived factor 1 (SDF-1) in CAFs, regulated by low levels of miR-1, enhance lung cancer cell A549 proliferation and drug resistance [84].

4.3. Tumor-Associated Angiogenesis

The development, differentiation, and homeostasis of endothelial cells are critical for tumor-associated vasculature for blood and nutrient supply. An angiogenic switch activates quiescent endothelial cells to form new blood vessels in addition to VEGF, angiopoietin, and FGF signals. Another possible pathway to study is the surface markers or gene profiles between normal and tumor endothelia. miR-494, in lung cancer, targets PTEN, and the subsequently activated Akt pathway promotes angiogenesis [54]. Our previous study showed that hypoxia-induced exosomal miR-23a downregulates prolyl hydroxylase 1/2 (PHD 1/2) and accumulates hypoxia-inducible factor-1α (HIF-1α) in endothelia [55]. One study suggested that angiogenesis-related miRs were altered, including miR-21, miR-106a, miR-126, miR-155, miR-182, miR-210, and miR-424, but these observations have not been validated [85].

4.4. The Epithelial-To-Mesenchymal Transition

The extracellular matrix (ECM) participates in steps involved in cancer metastasis. Many ECM proteins are dysregulated during the progression of cancer, both biochemically and biomechanically. Many ECM proteins are associated with the induction of epithelial-to-mesenchymal transition (EMT) [86,87]. EMT enables epithelial cancer cells to invade and metastasize [87]. There are several microRNAs regulating EMT-related genes. miR-149 targets Forkhead box M1 (FOXM1) to suppress EMT in A549 [88]. miR-33a targets Twist1 and inhibits EMT-related metastasis in NSCLC. Low levels of miR-33a carry a worse prognosis [89].

Taken together, miRs play roles in TME by controlling cancer cellular behaviors and dysregulating their interaction with surrounding cells. Investigating the relationships between cancer cells and their microenvironment is necessary in lung cancer treatment.

                                                                                                                                                        "

Another critical and very exciting aspect of the recent studies is the role of miRNAs downstream of specific driver mutations or signaling cascades. Even though authors have mentioned examples in the section on molecular therapy, perhaps a more scientific strategy would be to first describe the mutations or signaling cascades being targeted, then the established role of miRNAs in that pathway, and lastly the potential use of that information in clinic.

As suggested, we have expanded the spectra of driver mutations such as EGFR, ALK, ROS and KRAS. Their downstream signaling cascades and their relevant miRs are also included in this article (line 87-124)

Sustained cell proliferation and unsuppressed cell growth are thought to be the fundamental characteristics of cancer. Several genes and proteins are involved in this process, especially some kinases and kinases receptors [17]. By far, the epidermal growth factor receptor (EGFR) signaling pathway is the best-known example in lung cancer. Ligands such as epidermal growth factor (EGF) or transforming growth factor (TGF)-a bind to EGFR, resulting in its activation and receptor transphosphorylation. This process leads to two further major signaling pathways, Ras/Raf/MEK/ERK and PI3K/Akt/mTOR, and further enhances cell proliferation and cell cycle progression [18]. miRNAs that directly target EGFR are miR-7, miR-27a-3p, miR-30, miR-34, miR-128, miR-133, miR-134, miR-145, miR-146, miR-149, miR-218, and miR-542-5p [17,19–21].

Echinoderm microtubule-associated protein-like 4 (EML4) and anaplastic lymphoma kinase (ALK) fusion protein, which also initiate Ras/Raf/MEK/ERK and PI3K/Akt/mTOR pathways, are emerging therapeutic targets in NSCLC [22]. miR-96 was reported to be a post-transcriptional suppressor of ALK in cell models by Vishwamitra et al. [23].

ROS proto-oncogene 1 receptor tyrosine kinase (ROS1) is the third actionable driver mutation gene after EGFR and ALK, with available targeting therapeutics [24]. Similar to other receptor tyrosine kinases, ROS1 promotes cell survival and proliferation through activating downstream signaling pathways, such as SHP-1/SHP-2, JAK/STAT, PI3K/Akt/mTOR, and MEK/ERK [25]. miR-760 was found to reduce cell proliferation by suppressing ROS1 expression in NSCLC cell lines [26].

KRAS, a member of the Ras protein family, is a common downstream reactor of the previously mentioned receptor tyrosine kinases, e.g., EGFR, ALK, and ROS [27]. KRAS further initiates the Ras/Raf/MEK/ERK pathway, and it is estimated that 15% of lung cancer patients harbor KRAS mutations [28]. The let-7 family was reported to be able to repress KRAS and slow the proliferation of lung cancer cells, both in cell lines and in mouse models [29,30]. Seviour and co-workers reported that miR-193a-3p directly targets KRAS and inhibits KRAS-mutated lung tumor growth in vivo [31]. miR-181a-5p was also observed to downregulate KRAS, inhibiting the proliferation and migration of A549 cells [32]. Besides this, miR-148a-3p was shown to inhibit NSCLC cell proliferation, in vitro, via the suppression of SOS2, a molecule upstream of the Ras signaling pathway [33]. Jiang et al. also reported that miR-1258 may target GRB2 and inhibit Grb2, the upstream protein required for downstream Ras activation, downregulating the MEK/ERK pathway in mice models [34]. In another downstream PI3K/Akt/mTOR pathway, miR-520a-3p was reported to be involved in vitro [35].

 Regarding the role of cigarette smoke in inducing lung cancer, perhaps addition of a section on the effect of cigarette smoke on lung microRNAs and their relationship with tumorigenesis is warranted.

As suggested, we added this topic form line 247-270 as “The Interaction of Cigarette Smoke (CS) and miRANs in Lung Cancer Tumorigensis”

3. The Interaction of Cigarette Smoke (CS) and miRNAs in Lung Cancer Tumorigenesis

Long-term exposure to CS, which contains more than 70 carcinogens, is a well-established risk factor of lung cancer [69,70]. With increasing awareness in the human population of the potential harm, cigarette consumption is decreasing worldwide. However, the incidence of lung cancer has not declined accordingly. Air pollution and particulate matter are other important risk factors of lung cancer.

CS is linked to lung cancer formation via genomic alteration and epigenomic aberrant expression [71,72]. Genetically, carcinogens in CS may induce mutations in critical growth regulatory genes, such as KRAS and TP53 [71].

In addition to genomic alteration, exposure to CS also induces epigenomic dysfunction. First, smokers have differential DNA methylation profiles, which may interfere with the normal genetic expression in the small airway epithelium [72]. Second, CS induces histone modifications [73]. Sundar et al. reported that CS causes site-specific post-translational histone modifications (PTMs) of histones H3 and H4, which may play a role in lung cancer development [74]. Third, CS may alter miRNA expression. Many deregulated miRNAs are found to be associated with tumor suppressor gene silencing or oncogene upregulation. Advani et al. reported examples of dysregulated miRNAs using microarray-based approaches and bioinformatic analyses. In their H292 cell line with CS exposure, the downregulation of miR-155-5p, that targets CEBPB, and upregulation of miR-21-5p, that targets PDCD4, were observed, respectively [75]. In a recent study, a sound mechanism was proposed using human bronchial epithelium cells which elucidated that CS may induce HIF-1α-dependent miR-21 upregulation, reducing PTEN levels and, then, activation of the Akt/NF-κB pathway, leading to malignant transformation [76]. Nicotine, one of the various constituents in CS, is not thought to be a mutagen. However, nicotine is reported to promote proliferation and metastasis by suppressing miR-99b and miR-192, and upregulating FGFR3 and RB1 levels, respectively, in NSCLC cell lines [77].

The review also lacks a description of the basic miRNA machinery in the cell (e.g dicer and Drosha) and the studies performed on the role of these molecular machines in cancer.

Thanks for your suggestions. We did not describe it. We have added this part as bellow.

The biogenesis of miRNAs from source miRNA genes is a multistep process. Transcribed by RNA polymerase II, the large miRNA precursors, called pri-miRNAs, are >100 nucleotides in length. The pri-miRNAs are subsequently processed, intranuclearly, by the RNase III enzyme, Drosha, and the double stranded RNA (dsRNA)-binding protein, Pasha (also known as DiGeorge Syndrome critical region gene 8, DGCR8) [8]. The product of this process is called pre-miRNA, with a length of ~70 nucleotides. The pre-miRNAs are then transported into cytoplasm by a RanGTP-dependent dsRNA-binding protein, exportin 5 [9].

In the cytoplasm, another RNase III enzyme, Dicer, processes the pre-miRNAs into the miRNA:miRNA duplex of ~22 nucleotides. Generally, one chain of the miRNA duplex will bind to a multiprotein complex, named RNA-induced silencing complex (RISC). The single stranded miRNA in RISC acts as a template that recognizes the complementary mRNA, and then negatively regulates mRNA expression either by direct mRNA degradation or by translational repression, depending on the complementarity of miRNA and the target mRNA [10-12].

Reviewer 2 Report

This a very well written and comprehensive review of miRNAs in lung cancer..The authors identified a number of miRNAs expressed in lung cancer, from the practical therapeutic point of view, it is not clear how to identify key miRNAs for inhibitor development. Also I would like the authors to expand on the potential limitation of inhibitors development, such as in vivo toxicity, stability and delivery.

Author Response

Response to the Comments from Reviewer 2

This a very well written and comprehensive review of miRNAs in lung cancer. The authors identified a number of miRNAs expressed in lung cancer, from the practical therapeutic point of view, it is not clear how to identify key miRNAs for inhibitor development. Also I would like the authors to expand on the potential limitation of inhibitors development, such as in vivo toxicity, stability and delivery.

We appreciate the reviewer’s comments about this article. We have added the limitation of inhibitors development as bellow. 

Currently, the delivery system of miRNAs may be roughly divided into viral- or non-viral-based systems. Retroviruses, adenoviruses, and adeno-associated viruses are representative viral vectors. They have high transfection efficiency for delivering miRNAs. The viral vectors also have limited replication ability. However, there are some problems regarding their usage; for instance, toxin production, immunogenicity that induces inflammatory responses, and mutations caused by the inserted sequence [175]. Therefore, the viral-based delivery of miRNAs is a less-favored approach, clinically [176].

Non-viral systems, such as lipoplex or polymer-based methods, demonstrate lower toxicity and immunogenicity. Although they are much safer, their main disadvantage is the insufficient miRNA delivery rate. Among these, the most promising methods used locked nucleic acid (LNA) and liposomal nanoparticles of miRNA mimics, which changed the fate regarding rapid degradation of miRNAs in biofluid, and increased the delivery efficiency in vivo [176]. A phase II clinical trial using LNA-anti-miR (miravirsen or SPC3649; Santaris Pharma, Denmark) to target miR-122, for treating chronic hepatitis C virus infection, was completed with good tolerance and significant antiviral activity [177,178].

Due to the non-specific targets of miRNAs, unwanted off-target effects ensue if the delivered miRNA mimics, or anti-miRNAs, target unintended mRNAs. Besides this, the exogenous miRNA will compete with miRISC and endogenous miRNAs, interfering with normal regulation [179]. Beyond the off-target effects, an alteration of immune response is observed. For example, a grade 4 cytokine release syndrome was observed in a participant in the phase I clinical trial of MRX34, causing the premature closure of this trial [17].

Round  2

Reviewer 1 Report

2nd review IJMS Wu et al

1-In response to our comments authors had modified the manuscript and added some sections to the manuscript. This effort has expanded the size and scope of the paper. However, unfortunately the manuscript still contains innumerable errors in conveying the message and has to be improved before being ready for publication

I have included an example of editing needed below:

Page 1

-line 40: should be ‘A major concern with the LDCT screening method is its high false positive rate leading to overdiagnosis, high rates of radiation exposure, …’

Line 41:  LDCT does not induce overdiagnosis. The use of LDCT as a screening method leads to overdiagnosis

Line 43 urgently needed.

Line 44 the prognosis can not be unsatisfactory. Prognosis remains poor.

Line 44 should be:  'Patient with lung cancer…advanced stage, and more than …'

Line 50:  MicroRNAs can do both translational inhibition and degradation on the same RNA. So one can not use OR in this sentence

Line 52: ..miRNAs 'alter', not 'altered'

Line 53: lin4 is not the first miRNA. it is the first noncoding RNA that was identified as miRNA.

Line 53 was identified IN (not from)

Line 55: the sentence 'Since then,…' lacks a verb

Line 57: Source miRNA gene is not a valid biological term. Instead use miRNA genomic loci.

Line 57: sentence needs revision. Should be ‘pri-miRNAs are large (more than 100 nt) RNAs that are transcribed by RNA polymerase II and subsequently processed by…

Correction of all the errors is beyond the capacity of the reviewer. The high number of errors and misrepresentations makes the manuscript unfit for the audience of IJMS and necessitates a more rigorous editing and revision by the authors and a language editor.

2-It is not clear why authors do not refer to tumor endothelial cells as major components of the tumor microenvironment. Angiogenesis is not the only function of the tumor endothelial cells.  Similarly tumor epithelial cells deserve their subtitle.

Author Response

Response to the Comments from Reviewer 1

*2nd review IJMS Wu et al*

The authors would like to thank the reviewers’ thorough efforts to improve quality of this paper. The authors are not native English speaker. We have tried twice English Editing from the MDPI English Editing service and a native English speaker who is qualified to serve English editing in Taiwan. We do apologize for our poor English ability. However, we try our best to improve the quality of this article. We carefully checked every grammar errors and sentences, and did our best to avoid any misleading message to the readers. The errors were corrected point by point.  

1. In response to our comments authors had modified the manuscript and added some sections to the manuscript. This effort has expanded the size and scope of the paper. However, unfortunately the manuscript still contains innumerable errors in conveying the message and has to be improved before being ready for publication

I have included an example of editing needed below:

Page 1

-line 40: should be ‘A major concern with the LDCT screening method is its high false positive rate leading to overdiagnosis, high rates of radiation exposure, …’

We have corrected it into “A major concern with the LDCT screening method is its high false-positive rate, which leads to overdiagnosis, unnecessary radiation exposure, economic burden and patient anxiety [3].”

Line 41: LDCT does not induce overdiagnosis. The use of LDCT as a screening method leads to overdiagnosis

We have corrected it into “A major concern with the LDCT screening method is its high false-positive rate, which leads to overdiagnosis, unnecessary radiation exposure, economic burden and patient anxiety [3].

Line 43 urgently needed.

We modified it into “Thus, the development of more accurate methods for the diagnosis of lung cancer is urgently needed.”

Line 44 the prognosis can not be unsatisfactory. Prognosis remains poor.

We have rewritten into “Despite emerging advances in early diagnosis and novel targeted agents, the prognosis of lung cancer remains poor.”

Line 44 should be:  'Patient with lung cancer…advanced stage, and more than …'

It was modified as “Patients with lung cancer are often diagnosed at advanced stages, and usually more than 60% of patients are at stages III or IV before treatment [4].”

Line 50:  MicroRNAs can do both translational inhibition and degradation on the same RNA. So one can not use OR in this sentence

We have modified it into “MicroRNAs (miRNA) are a family of small noncoding RNAs (21–25 nucleotides) that can inhibit messenger RNA (mRNA) translation and promote mRNA degradation by base pairing to complementary sites of target mRNAs.

Line 52: ..miRNAs 'alter', not 'altered'

As suggested, we have changed it into “Through this mechanism, miRNAs alter gene expression post-transcriptionally.

Line 53: lin4 is not the first miRNA. it is the first noncoding RNA that was identified as miRNA.

We modified it into “The first noncoding RNA, lin-4, was identified as miRNA in Caenorhabditis elegans in 1993 [6];

Line 53 was identified IN (not from)

We changed it into “over 2500 mature miRNAs (from 1188 miRNA precursors) have been identified in the miRBase

Line 55: the sentence 'Since then,…' lacks a verb

We modified the sentence as “while now, over 2500 mature miRNAs (from 1188 miRNA precursors) have been identified in the miRBase

Line 57: Source miRNA gene is not a valid biological term. Instead use miRNA genomic loci.

We modified it as “The biogenesis of miRNAs from miRNA genomic loci is a multistep process.

Line 57: sentence needs revision. Should be ‘pri-miRNAs are large (more than 100 nt) RNAs that are transcribed by RNA polymerase II and subsequently processed by…

We have modified it as “, pri-miRNAs, are large miRNAs (>100 nucleotides in length) transcribed by RNA polymerase II and subsequently processed,

Correction of all the errors is beyond the capacity of the reviewer. The high number of errors and misrepresentations makes the manuscript unfit for the audience of IJMS and necessitates a more rigorous editing and revision by the authors and a language editor.

We, authors, thanks the efforts by the reviewers and are sorry for the poor English writing and errors we have made. We have tried our best to improve the content. Also, we have another native English speaker to modify our article. The attached files are the English Editing certificates at final page of this review article.  

2. It is not clear why authors do not refer to tumor endothelial cells as major components of the tumor microenvironment. Angiogenesis is not the only function of the tumor endothelial cells.  Similarly tumor epithelial cells deserve their subtitle.

Thanks for the reviewer’s suggestion. We have added the subtitle and rewritten it as

4.3. Tumor Endothelial Cells and Tumor-Associated Angiogenesis

The development, differentiation, and homeostasis of endothelial cells are critical for tumor-associated vasculature for blood and nutrient supply. Tumor endothelial cells, lining tumor blood vessels, exhibit altered phenotypes compared with normal counterparts. They demonstrate activated proliferation and migration, and upregulate several angiogenesis-related genes [82]. Some endothelial-specific miRNAs have been involved in various aspects of angiogenesis as proliferation, migration and morphogenesis of endothelial cells [83]. In one study, several miRNAs including miR-142-3p, miR-143-3p, etc. derived from inhibited lung cancer extracellular vesicles. Our previous study showed that hypoxia-induced exosomal miR-23a downregulates prolyl hydroxylase 1/2 (PHD 1/2) and accumulates hypoxia-inducible factor-1α (HIF-1α) in endothelia [53].

An angiogenic switch activates quiescent endothelial cells to form new blood vessels in addition to VEGF, angiopoietin, and FGF signals. Another possible pathway to study is the surface markers or gene profiles between normal and tumor endothelia. miR-494, in lung cancer, targets PTEN, and the subsequently activated Akt pathway promotes angiogenesis [52]. One study suggested that angiogenesis-related miRs were altered, including miR-21, miR-106a, miR-126, miR-155, miR-182, miR-210, and miR-424, but these observations have not been validated [84].
